# Albumin in Normovolemic Fluid Management for Severe Traumatic Brain Injury: Controversies and Research Gaps

**DOI:** 10.3390/jcm13185452

**Published:** 2024-09-13

**Authors:** Christian J. Wiedermann

**Affiliations:** 1Institute of General Practice and Public Health, Claudiana—College of Health Professions, 39100 Bolzano, Italy; christian.wiedermann@am-mg.claudiana.bz.it; 2Department of Public Health, Medical Decision Making and Health Technology Assessment, UMIT TIROL—Private University for Health Sciences and Health Technology, 6060 Hall, Austria

**Keywords:** traumatic brain injury, serum albumin, normovolemia, fluid management, acute kidney injury, Lund concept, small-volume resuscitation, hyperoncotic albumin

## Abstract

Traumatic brain injury (TBI) is a significant public health issue characterized by high mortality rates and long-term complications. This commentary examines the controversial role of the use of albumin in the fluid management of patients with severe TBI. Despite its physiological benefits, the clinical use of albumin remains controversial due to the fact that various studies have yielded mixed results. Serum albumin is important for maintaining normovolemia, primarily through its contribution to colloid osmotic pressure, which helps to retain fluid in the circulatory system. This review highlights the existing evidence, examines inconsistencies in guideline recommendations, and suggests future research directions to clarify the efficacy and safety of the use of albumin in maintaining normovolemia in patients with TBI. The review also discusses the potential benefits of small-volume resuscitation strategies for the management of acute kidney injury in TBI patients, drawing parallels with the management of septic acute kidney injury. The need for further well-designed randomized controlled trials and ethical considerations in studies regarding the use of hyperoncotic albumin in TBI management is emphasized.

## 1. Introduction

Traumatic brain injury (TBI) is a significant public health concern characterized by high mortality rates and long-term complications that profoundly impact survivors’ quality of life [1]. TBI encompasses a spectrum of injuries resulting from external mechanical forces that lead to primary and secondary brain damages. The pathophysiology of TBI is complex and involves immediate physical injury followed by a cascade of inflammatory responses that exacerbate neuronal damage and cerebral edema [2]. Effective management of TBI involves maintaining adequate cerebral perfusion pressure (CPP) and controlling intracranial pressure (ICP) [3]. Given the intricate interplay of these factors, precise hemodynamic and fluid management is the cornerstone of TBI care to prevent secondary injuries and optimize patient outcomes [4].

Preventing fluctuations in blood pressure is important because both low and high systolic blood pressure can be detrimental. Judicious crystalloid administration is preferred for euvolemia, whereas vasoactive agents are preferred for fluid non-responders [5]. Early vasopressor use in TBI is independently associated with increased mortality, irrespective of the patient’s fluid volume status [6]. There is a risk of severe TBI leading to hypovolemia soon after the trauma, even in the absence of extracranial bleeding [7,8]. Hemodynamic compromise in TBI can result from brain injury-associated shock, characterized by a catecholamine storm that leads to ventricular dysfunction and vasoplegia [9]. This hemodynamic instability complicates the management of TBI, as it can be clinically indistinguishable from hemorrhagic shock, posing challenges for appropriate diagnosis and treatment. 

Guidelines emphasize the importance of fluid management in acute brain injury to maintain normovolemia, or normal blood volume, and to minimize the risk of complications related to fluid imbalance (Figure 1) [10]. The primary goal of conventional fluid management in patients with TBI is to maintain a stable hemodynamic condition through the use of both invasive and noninvasive monitoring methods to track arterial blood pressure, fluid balance, and urinary output [3]. A study of patients with subarachnoid hemorrhage, focusing on the relationship between volemic status, as monitored through invasive hemodynamic techniques, and brain tissue oxygenation, found that brain tissue oxygenation levels did not vary significantly between hypovolemic and euvolemic states, showing that factors other than fluid volume status influence brain oxygenation [11].

Traditionally, albumin, a colloid with strong oncotic properties, has been used in fluid resuscitation because of its capacity to expand plasma volume and maintain intravascular pressure. However, its application in TBI is contentious, as clinical trials have yielded mixed results. Some studies indicate that albumin may mitigate cerebral edema [13] and stabilize hemodynamics [14], while others, such as the SAFE-TBI study [15], have reported increased mortality associated with its use, potentially due to the exacerbation of ICP. These disparate outcomes have engendered a significant discourse in the field, with concerns about the balance between albumin’s theoretical advantages and the practical risks it may present in TBI management.

The aim of this commentary is to discuss the controversial role of albumin in fluid management of patients with severe TBI. Despite the established physiological role of serum albumin, the use of albumin infusion in clinical practice, particularly in TBI management, remains controversial owing to the mixed outcomes from various studies. This article reviews the existing evidence, highlights the inconsistencies in clinical guidelines, and proposes directions for future research to clarify the efficacy and safety of albumin in maintaining normovolemia in patients with TBI.

## 2. The Role of Serum Albumin in TBI

Previous studies have shown that decreased serum albumin levels and increased C-reactive protein/serum albumin ratios are independent risk factors for mortality in patients with traumatic brain injury (TBI). Furthermore, these indicators exhibit predictive efficacy for short- and long-term outcomes in both pediatric and adult populations [16,17,18,19,20,21]. Experimental [22,23] and clinical [13] studies have shown that albumin infusion can reduce cerebral edema. Serum albumin plays an important role in the maintenance of normovolemia, primarily through its contribution to colloid osmotic pressure, which helps retain water in the circulatory system and prevents fluid leakage into the tissues [24]. In the clinical setting, albumin can be administered to effectively expand plasma volume by drawing water from the interstitial space into the bloodstream [25,26,27]. In addition, albumin acts as a carrier protein for various substances, stabilizing them in the bloodstream and contributing to the buffering capacity of the blood to maintain the pH balance. Its antioxidant properties help neutralize free radicals and protect blood vessels and tissues, which are important for fluid balance. Albumin also binds to toxins and waste products, helping to detoxify and remove them, preventing tissue damage and supporting fluid regulation [28]. These functions may be particularly important in severe TBI as an inflammatory condition [2], where precise fluid management prevents complications and ensures hemodynamic stability [4].

The advantages of albumin include the preservation of normal blood volume and decreased cerebral edema; however, its clinical application is not without peril. Certain investigations have implied that administering albumin to TBI patients could lead to an increase in ICP [15]. In the following sections of this article, these hazards will be explored, unravelling the clinical controversies and evaluating the balance between the benefits and potential adverse effects.

## 3. Fluid Management Strategies in TBI

Crystalloid infusions are the preferred option for the maintenance and resuscitation of patients with traumatic brain injury (TBI). However, in the past, hypotonic and albumin solutions were not recommended. It is crucial to emphasize that hypertonic saline solutions, when utilized as maintenance or resuscitation fluids, do not provide any advantage over saline or isotonic balanced solutions [3]. The benefits and drawbacks of fluid loading in comparison with the use of vasoactive drugs, as well as the impact of the selection of specific vasoactive drugs, are unclear and uncertain. The use of vasopressors to maintain CPP early in the resuscitation process may conceal the presence of under-resuscitation [5]. It is essential to assess the volume status prior to the initiation of vasopressors and periodically thereafter, as fluid overload can worsen outcomes, but hypovolemia should also be avoided [4].

### 3.1. The Shift towards Normovolemia

The observational study conducted by the CENTER-TBI and OzENTER-TBI collaboration groups emphasizes the shift in fluid management practices for critically ill TBI patients towards achieving normovolemia, defined as a neutral or net zero fluid balance [29]. The European Society of Intensive Care Medicine (ESICM) consensus on fluid therapy advocates for this approach, using arterial blood pressure and fluid balance as key metrics to guide fluid administration while avoiding a negative fluid balance [4]. Positive fluid balance, indicative of hypervolemia, is associated with higher ICU mortality and worse functional outcomes at six months post-injury, highlighting the importance of maintaining normovolemia to optimize clinical outcomes and reduce complications of TBI [29,30]. Based on observational evidence, normovolemia should be the goal in order to avoid secondary brain injury due to hypovolemia or hypervolemia [5]. Interruption of the blood–brain barrier (BBB) can lead to extravasation of small amounts of fluid into the extravascular interstitium of the brain, which can impact brain compliance. In these situations, it is critical to preserve euvolemia by employing isotonic normal saline rather than hypotonic solutions, such as lactated Ringer’s or 4–5% albumin, as the latter can exacerbate cerebral edema [31].

#### Normovolemia and TBI-Associated Acute Kidney Injury

As per De Vlieger and Meyfroidt [32], it is essential to accurately assess the volume status of patients with TBI, as studies have demonstrated that hypovolemia is linked to an increased risk of mortality [33]. Moreover, patients with TBI who were admitted to the ICU in a country with a high incidence of acute kidney injury (AKI) displayed a higher fluid balance three days after admission [34]. Observational studies on a large scale have suggested that fluid accumulation is prevalent in individuals with AKI. This phenomenon begins prior to the emergence of AKI and persists in worsening following the onset of the condition [35]. Robba et al. [34] found that AKI occurs in approximately 10% of TBI patients within the first week of ICU admission, with a median onset of 2 days. Significant risk factors for AKI include a history of renal issues, insulin-dependent diabetes, hypernatremia, and the use of osmotic therapy. AKI was associated with a longer ICU stay, higher mortality, and worse outcomes on the Extended Glasgow Outcome Scale at 6 months post-injury. The study emphasized the importance of maintaining euvolemia using isotonic solutions, e.g., normal saline, to prevent complications such as cerebral edema [34]. Moreover, it has been discovered that the accumulation of fluids in critically ill patients is independently associated with both ICU mortality and AKI. Additionally, it this accumulation is also linked to a decreased probability of renal recovery [36].

The pilot trial for the Restrictive Fluid Management Versus Usual Care in Acute Kidney Injury (REVERSE-AKI) project has recently demonstrated that a combination of restrictive fluid management and diuretics for patients with early AKI is associated with a reduced risk of worsening renal function [37]. The management of AKI in TBI may benefit from small-volume resuscitation strategies, as evidenced by studies in patients with sepsis, where such approaches have been shown to be beneficial [38]. The efficacy of this method in individuals with TBI has yet to be established. Nevertheless, it has been observed that the use of transthoracic ultrasound-guided fluid resuscitation at the time of emergency department admission can result in a reduction in the volume of fluids administered, which was associated with a lower mortality rate in a subset of 72 TBI patients when compared with the results for patients receiving conventional care [39]. Various techniques are currently available to evaluate intravascular volume levels, which can aid in determining the appropriate administration of fluids [40].

### 3.2. The Lund Concept Approach to Normovolemia

The approach to maintaining normovolemia in the hemodynamic management of TBI [13] differs between the current treatment recommendations of the Brain Trauma Foundation (BTF) guidelines [14,41] and the Lund concept (LC) [42].

The BTF guidelines provide minimal guidance regarding fluid management, emphasizing the importance of balancing fluid therapy to avoid hypovolemia and fluid overload. The primary goal is to maintain adequate blood pressure without causing excessive fluid accumulation [41].

The LC places greater emphasis on precise intravascular volume management to maintain normovolemia. It advocates the use of isotonic 20% albumin in combination with crystalloids, dosed to maintain serum albumin levels of 32 g/L or above, to limit the total volume of fluid required, thereby minimizing the risk of cerebral edema. The LC strategy attempts to avoid the use of vasopressors and inotropes and recommends early and continuous administration of beta-blockers and alpha-2 agonist therapy to control blood pressure and reduce adrenergic stress [42].

The LC emphasizes precise intravascular volume management and has demonstrated potential benefits, but it remains a topic of discussion within the medical community. Critics argue that the LC’s avoidance of vasopressors and reliance on albumin may not be suitable for all patients with TBI, particularly in cases where maintaining blood pressure is difficult [43]. Furthermore, the evidence supporting the LC primarily comes from observational studies and smaller trials, leading to concerns about the generalizability of these findings [44]. The lack of large, randomized controlled trials (RCTs) diminishes the robustness of the LC’s recommendations, and thus, it is not universally accepted in clinical practice.

#### 3.2.1. Rationale for Using Albumin in the Lund Concept

Table 1 summarizes the biological plausibility of using albumin for normovolemic fluid management.

The primary advantage of albumin is its higher oncotic pressure compared to that of crystalloids. This characteristic allows albumin to maintain intravascular volume more effectively [27,51], which is critical in patients with TBI, in which adequate cerebral perfusion and the prevention of cerebral edema are paramount. The blood–brain barrier (BBB) is often compromised in brain injury, permitting electrolytes and smaller molecules to pass through. Albumin, a larger molecule, is less prone to leakage through the disrupted BBB, thereby helping to sustain a stable intravascular volume without significantly contributing to brain swelling [52].

In the Albumin Italian Outcome Sepsis (ALBIOS) study, albumin dosing to maintain serum levels of 30 g/L enabled a lower net fluid balance and higher mean arterial pressure than did saline dosing, despite the administration of similar total fluid amounts [46]. Albumin reduces the total fluid volume required to maintain normovolemia [48,49]. This may be particularly beneficial, as it lowers the risk of fluid overload, which can worsen cerebral edema due to the disrupted BBB in TBI patients.

Administration of hyperoncotic albumin improves fluid removal during renal replacement therapy by promoting plasma replenishment and preventing intradialytic hypotension. In a randomized crossover study, patients with hypoalbuminemia undergoing intermittent hemodialysis experienced significantly fewer episodes of hypotension and better fluid removal when receiving albumin than when receiving saline [50]. This supports the hypothesis that the normalization of serum albumin levels in patients with TBI may improve fluid management.

Albumin works by drawing interstitial fluid back into the bloodstream [25,26,27], thus increasing the effective circulating volume. This mechanism is especially useful in counteracting the hypovolemic state observed in severe TBI caused by increased capillary permeability and subsequent fluid loss. Experimental studies in animals have demonstrated that albumin can reduce brain edema and enhance systemic microcirculation and hemodynamics. Despite some criticism and mixed results in clinical studies, the overall findings support the use of albumin in achieving better control over intravascular volume and reducing adverse outcomes in patients with TBI.

The use of hyperoncotic albumin in normovolemia management is further supported by evidence suggesting that hypoalbuminemia moderates the volume expansion effect of albumin [47]. This indicates that maintaining serum albumin levels at ≥ 32 g/L may enable effective plasma expansion with smaller infusion volumes, enhancing hemodynamic stability while minimizing the risk of cerebral edema.

#### 3.2.2. Clinical Evidence for the Lund Concept

Clinical evidence and support for the LC approach, although sometimes controversial, generally indicate that albumin use can lead to better outcomes by ensuring more stable hemodynamics and reducing complications such as acute respiratory distress syndrome associated with fluid overload. A meta-analysis of four controlled clinical trials found that the use of 20–25% albumin solution as part of the LC reduced mortality compared with alternative treatments. Among 165 patients treated with albumin, 24 (14.5%) died, while 59 of the 155 control patients (38.1%) did not survive, but the trials were at high risk of bias [44].

### 3.3. Controversies and Criticisms of SAFE-TBI

Studies of patients with severe TBI have shown that plasma albumin levels decrease significantly after trauma [16,53,54], and low levels predict poor outcomes [15]. Experimental studies [22,23] and clinical trials [13] have shown that albumin can reduce cerebral edema. However, the SAFE-TBI trial in Australia and New Zealand reported worse outcomes for albumin than for saline, suggesting increased intracranial ICP as a possible cause [14]. This significant study, published in the New England Journal of Medicine, has exerted a profound impact on clinical practice guidelines. However, it is crucial to acknowledge that the trial’s findings were derived from a post-hoc analysis of a subset of participants, which has engendered concerns about the broader applicability of the results [55,56]. In particular, the use of a hypotonic albumin solution, which contrasts with the isotonic preparations suggested in other research, could have played a role in the higher ICP recorded [57]. This suggests that the oncotic properties, rather than the albumin molecule itself, may have played a critical role in the adverse outcomes [58,59]. Given these observations, the SAFE-TBI study alone cannot discredit the use of non-hypotonic albumin in severe TBI patients [60]. The SAFE-TBI study has had a significant influence on guidelines for TBI management, and its impact cannot be overestimated. Nevertheless, it is crucial to acknowledge the need for additional research to determine the circumstances under which albumin can be employed safely and effectively in TBI management.

Furthermore, it is essential to recognize the positive outcomes observed in other studies utilizing isotonic albumin preparations [42]. These findings suggest that, with appropriate formulation and clinical context, albumin may still hold potential benefits for plasma volume expansion in severe TBI patients.

## 4. Guideline Recommendations against the Use of Hyperoncotic Albumin in Severe TBI

According to Oddo et al. [4], not all concentrations of albumin solutions are recommended for fluid therapy in TBI due to the observed associated risks, including the potential for increased ICP and higher mortality rates. These guidelines are based on limited evidence supporting the benefits of albumin in this context and the significant risks observed in clinical trials [15]. Most recently, the ESICM guidelines on fluid resuscitation have recommended against the use of albumin solutions in patients with TBI [45]. The ESICM guidelines conclude that no robust data currently address the safety and efficacy of hyperoncotic (20–25%) human albumin solutions in patients with severe TBI [44].

In severe TBI, the BBB may become compromised, leading to increased capillary permeability. Albumin, a large colloidal molecule, may extravasate from the intravascular space into the interstitial space of the brain. Once in the interstitial space, albumin can draw water due to its osmotic properties, potentially worsening edema formation, and possibly affecting the inflammatory and reparative pathophysiology [61]. This theoretical mechanism raises concerns about the safety of albumin in TBI management, although direct experimental evidence supporting this hypothesis remains limited [62].

Due to the inconsistent recommendations on fluid management in TBI, clinicians must adopt an individualized approach to patient care. When considering albumin therapy, clinicians should consider factors such as the presence of AKI, baseline serum albumin levels, and the potential for increased ICP. Given the mixed evidence, particularly from studies like SAFE-TBI, clinicians should exercise caution and closely monitor both hemodynamics and brain-specific indicators. As further research is needed to provide clearer guidance, fluid management for patients with TBI should be tailored to meet their specific needs and risks.

## 5. Ongoing and Future Studies 

The WHO International Clinical Trials Registry Platform (ICTRP) is a global initiative providing a complete picture of clinical research. It unites clinical trials from different registries worldwide into a single searchable database [63]. 

On June 29, 2024, an advanced search on the WHO ICTRP, using “traumatic brain injury” as the condition term and “albumin OR fluid” as the intervention term, yielded 10 trials that were either recruiting or not recruiting. Among these trials, one investigated neuromodulation with cerebrolysin, one was a pharmacokinetic study of levetiracetam, one focused on decompression, and one was a long-term follow-up of patients with TBI conducted by the Karolinska Institute. Another study investigated the implementation of standard operating procedures for the care of patients with TBI, whereas another study investigated risk prediction using ischemia-modified serum albumin. In addition, one study focused on predicting fluid responsiveness.

Of the remaining three studies, two focused on normovolemia (ISRCTN23139643, registered in 2019, and NCT05983549, registered in 2023), and one studied the LC (ChiCTR1900026592). The LC trial, registered in 2019 in China, aimed to evaluate the efficacy of the LC in the management of TBI [64]. In the intervention group, hemoglobin and plasma albumin levels were maintained by infusing red blood cells to maintain hemoglobin at levels between 12.5 and 14 g/dL and administering 20–25% albumin to maintain albumin levels between 35 and 43 g/dL. ICP is primarily reduced by lowering the CPP. The control group followed a dehydration- and ICP-lowering program, based on hypertonic drugs such as mannitol. The principal aim of this study was to reduce the 28-day mortality rate. Other objectives included minimizing the length of the hospital stay, the time spent in the ICU, hospital expenses, mechanical ventilation time, the APACHE II score, the GCS score, the mannitol dosage, the 60-day mortality rate, and the 28- and 60-day Glasgow Outcome Scale (GOS) scores. Despite its registration more than five years ago, the study has not been updated, and attempts to contact the authors have been unsuccessful. A search of the EMBASE and PubMed databases for the names and affiliations of those responsible for the study did not reveal any publication of the results. The lack of updates and publications from the LC trial raises questions regarding its progress and transparency. The researchers involved should be encouraged to provide updates on their studies. This includes publishing interim results or providing a status report, even if the trial faces challenges.

Additional well-designed randomized controlled trials (RCTs) are needed to confirm the efficacy and safety of hyperoncotic albumin in the treatment of TBI. These studies should address the limitations of previous trials [44] and include comprehensive monitoring of all potential adverse effects [65]. Future research could examine the potential for utilizing iso- or hypertonic albumin solutions to correct hypoalbuminemia, while simultaneously closely monitoring hemodynamics and brain-specific parameters. Additionally, it is important to consider the presence of AKI when individualizing albumin therapy.

Emerging research on Long COVID suggests that its neurological effects may impede TBI recovery and influence responses to treatments such as albumin [66]. Future studies should examine the impact of Long COVID on brain injury outcomes, especially regarding fluid management, to ensure comprehensive care for this patient group. Furthermore, artificial intelligence offers considerable potential for enhancing the management of TBI and for research on albumin therapy. Specifically, it can predict patient responses, optimize fluid management protocols, and analyze extensive datasets from clinical trials to improve treatment strategies [67]. Collaboration between institutions and researchers specializing in TBI and neurocritical care can help design robust trials and pool data for meta-analyses to provide more definitive answers.

The use of isooncotic or hyperoncotic albumin in RCTs for TBI must adhere to strict ethical standards, owing to mixed evidence regarding its efficacy and safety. To ensure the highest ethical standards, investigators must openly disclose any potential risks associated with their research, including the possibility of exacerbating existing conditions such as ICP, as well as the development of other unfavorable consequences. It is essential that informed consent procedures explicitly detail the risks and uncertainties related to albumin treatment. Trial designs should incorporate rigorous monitoring strategies that allow for swift action in the event of adverse outcomes. Ethical review boards must actively scrutinize study protocols to prioritize patient safety, while simultaneously considering the broader implications of the research on future clinical guidelines and patient care standards. Ultimately, the potential benefits of the research must be deemed to outweigh the risks involved.

## 6. Conclusions

The management of TBI remains a challenging area of critical care, with a major impact on patient outcomes. Maintaining normal serum albumin levels in normovolemic fluid management offers potential benefits such as hemodynamic stability and reduced cerebral edema. Although the current evidence is mixed and controversial, there is a strong rationale for further clinical research. Future studies should focus on well-designed RCTs to evaluate the efficacy and safety of the use of hyperoncotic albumin in patients with TBI, with the inclusion of the comprehensive monitoring of adverse effects. Collaboration between researchers and institutions is essential to increase the robustness of these studies and develop evidence-based guidelines. Improving clinical practice and enhancing outcomes for patients with TBI is of paramount importance, making it essential for future research to elucidate the role of albumin in TBI management. It is imperative to verify the efficacy and safety of therapeutic methods to improve patient care, especially when considering the high stakes involved.

## Figures and Tables

**Figure 1 jcm-13-05452-f001:**
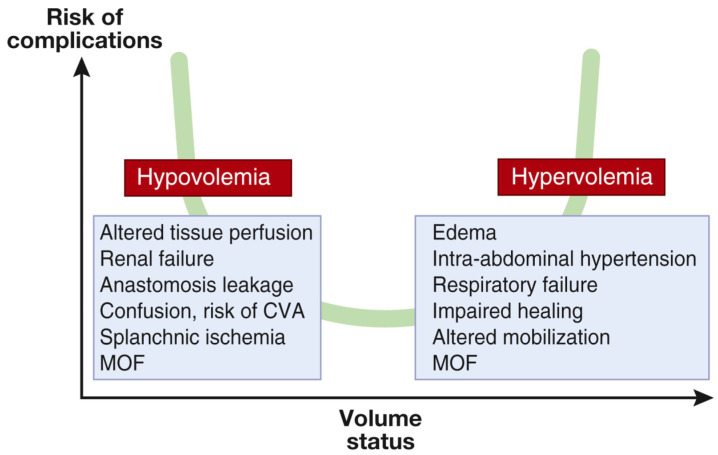
Relationship between blood volume and the risk of complications. Both hypo- and hypervolemia should be prevented, as each increases the risk of complications. CVA, cerebrovascular accident; MOF, multiple organ failure. From reference [12]. Copyright © 2019, International Society of Nephrology. Published by Elsevier Inc.

**Table 1 jcm-13-05452-t001:** Potential benefits of albumin in normovolemic fluid management for TBI.

Key Points	Details
Increased Oncotic Pressure	Maintains intravascular volume more effectively than crystalloids [45].
	Larger molecular size reduces leakage, sustaining stable intravascular volume [45].
	Draws interstitial fluid back into the bloodstream [28].
Anti-inflammation	Experimental and clinical evidence suggest albumin reduces brain edema and enhances microcirculation [13,22,23].
Improved Fluid Balance	In sepsis, serum albumin at 30 g/L led to a lower net fluid balance and higher arterial pressure compared to the results for saline [46].
	Reduces total fluid volume needed in hypoalbuminemia [47], lowering risk of fluid overload and cerebral edema.
	Maintaining serum albumin at the normal range enables effective plasma expansion with smaller volumes [48,49].
Hemodynamic Stabilization	Prevents episodic hypotension by promoting plasma replenishment [50].

## Data Availability

No new data were created.

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
