# Peer review of "Albumin in Normovolemic Fluid Management for Severe Traumatic Brain Injury: Controversies and Research Gaps"

_jcm, 2024, doi:10.3390/jcm13185452_

Round 1

Reviewer 1 Report

Comments and Suggestions for Authors

The commentary titled "Albumin in Normovolemic Fluid Management for Severe Traumatic Brain Injury: Controversies and Research Gaps" by Christian J. Wiedermann explores the ongoing debate regarding the use of albumin in fluid management for patients with severe traumatic brain injury (TBI). Despite the theoretical benefits of albumin in maintaining normovolemia, the commentary highlights inconsistent findings across clinical studies and gaps in research, particularly regarding its safety and efficacy.

Comments:

  1. Introduction (Lines 24-68): The introduction adequately outlines the complexities of TBI management but could benefit from a more focused discussion on the specific controversies surrounding albumin use (why albumin specifically).
  2. Discussion of Serum Albumin (Lines 69-86): The section discusses the physiological roles of albumin, but it fails to critically engage with the limitations and risks associated with its clinical use, particularly the potential for increased intracranial pressure (ICP).
  3. Lund Concept (Lines 142-204): While the benefits of albumin within the Lund Concept are discussed, the commentary does not sufficiently address the lack of consensus in the medical community and the criticisms of this approach.
  4. Controversies and Criticisms (Lines 205-224): The commentary references the SAFE-TBI trial, which reported worse outcomes with albumin use, but the discussion is somewhat dismissive of these findings, attributing them to study design flaws without fully considering their implications. The commentary may overstate the potential of albumin.
  5. General Feedback:
    • The abstract briefly summarizes the commentary's focus and the title is appropriate.
    • The commentary could be strengthened by a more balanced discussion acknowledging the criticisms of albumin use in TBI and providing a clearer rationale for future research directions.
    • The conclusion appropriately calls for further research.
Comments on the Quality of English Language

The English language is fine

Reviewer 2 Report

Comments and Suggestions for Authors

In general, the paper is well-written and scientifically sound, with minor areas that could be refined for clarity and structure. Strengthening the neutral presentation of controversial studies and ensuring clear, concise language throughout will enhance the paper’s impact.

Q1: Abstract: The abstract is well-structured, but it could be slightly more concise. The final sentence could be rephrased to summarize the paper's conclusions more clearly.

Q2: line 10:This commentary examines the controversial role of albumin in fluid management of patients with severe TBI.

 Consider rephrasing to “This commentary examines the controversial role of albumin in the fluid management of patients with severe TBI.”

Q3: line 11: Despite the physiological benefits of albumin, its administration in clinical practice remains controversial owing to mixed results from clinical studies.

 could be clearer if rewritten as, “Despite its physiological benefits, the clinical use of albumin remains controversial due to mixed results from studies.”

Q4: Introduction: The introduction provides a good background but could benefit from a brief summary of the controversy surrounding albumin use early on, setting the stage for the discussion.

Q5: Conclusion: The conclusion is well-summarized, but it could benefit from a stronger final statement regarding the need for future research or a clear take-home message.

Q6: It seems that this manuscript presented both sides of the controversy well, but it leaned slightly more towards advocating for albumin use. It might benefit from a more neutral tone when discussing the SAFE-TBI study to avoid perceived bias.

Q7: Discussion: Although the paper effectively discusses controversies, however, I would suggest that the authors consider elaborating more on the specific mechanisms by which albumin could potentially exacerbate cerebral edema, as this is a central concern in its use for TBI patients.

Q8: The information in Table 1 is critical. I would suggest that the authors should add citing references in the end of each line. 

Comments on the Quality of English Language

Minor English editing is indicative.

Reviewer 3 Report

Comments and Suggestions for Authors

Comments and suggestions:

  1. Long COVID omission: The author does not mention Long COVID or its potential influence on brain injury outcomes. This is a significant oversight, as emerging research suggests that Long COVID can have neurological impacts that may complicate TBI recovery or alter how patients respond to treatments like albumin. Including a brief discussion on this topic would have made the paper more comprehensive and timely.
  2. Artificial Intelligence (AI) as a future avenue: The manuscript does not discuss the potential role of AI in TBI management or albumin therapy research. This is a missed opportunity, as AI could be valuable for predicting patient responses to albumin therapy, optimizing fluid management protocols, or analyzing large datasets from clinical trials. A section on future directions should have included AI as a promising avenue for advancing this field.
  3. Structure and organization: The paper is well-structured overall, with clear sections addressing different aspects of albumin use in TBI. However, some sections, particularly those discussing clinical evidence and ongoing studies, could be more concisely presented.
  4. Literature review: The author provides a comprehensive review of existing literature on the topic. They effectively present both supporting and contradictory evidence regarding albumin use in TBI, which gives a balanced view of the current state of research.
  5. Critical analysis of SAFE-TBI trial: The author provides a good critical analysis of the SAFE-TBI trial, highlighting its limitations and the controversies surrounding its results. This adds depth to the discussion and demonstrates the complexity of interpreting clinical trial data in this field.
  6. Guideline recommendations: The paper effectively presents the current guideline recommendations against hyperoncotic albumin use in severe TBI, which is important for clinical practice.
  7. Future research directions: While the author discusses the need for additional well-designed randomized controlled trials, he could have provided more specific suggestions for study designs or research questions that need to be addressed.
  8. Practical implications: The paper could benefit from a more explicit discussion of the practical implications of the current evidence for clinicians managing TBI patients.
  9. International perspective: The author includes information from various countries and international trials, which provides a good global perspective on the topic.
  10. Ethical considerations: The brief mention of ethical considerations in conducting trials with hyperoncotic albumin is important, but this section could be expanded to provide more guidance for future researchers.

Overall, this is a comprehensive and well-researched manuscript that provides a balanced view of the current state of albumin use in TBI management. However, addressing the omissions regarding Long COVID and AI, and providing more specific future research directions would significantly enhance its value to the field.

Round 2

Reviewer 2 Report

Comments and Suggestions for Authors

It seems that the authors have made a great effort in revising their manuscript according to the reviewer's suggestions. It looks very good. I would suggest this work to be accepted.

Reviewer 3 Report

Comments and Suggestions for Authors

The author responded to my comments very well. Thank you.